# Practical Bayesian Algorithm Execution via Posterior Sampling

## Abstract

We consider the *Bayesian algorithm execution* framework, where the goal is to select points for evaluating an expensive function to best infer a property of interest. By making the key observation that the property of interest for many tasks is a *target set* of points defined in terms of the function, we derive a simple yet effective and scalable posterior sampling algorithm, termed PS-BAX. Our approach addresses a broad range of problems, including many optimization variants and level-set estimation. Experiments across a diverse set of tasks show that PS-BAX achieves competitive performance against standard baselines, while being significantly faster, simpler to implement, and easily parallelizable. In addition, we show that PS-BAX is asymptotically consistent under mild regularity conditions. Consequently, our work yields new insights into posterior sampling, broadening its application scope and providing a strong baseline for future exploration.

## 1   Introduction

Many real-world problems can be cast as estimating a property of a black-box function with expensive evaluations. Bayesian optimization [1] has focused on the case where the property of interest is the function's global optimum. Similarly, level set estimation [2] deals with the problem of estimating the subset of points above a user-specified threshold.

More generally, it is often of interest to compute a property of the function determined by the output of a *base algorithm*. However, the base algorithm usually requires a large number of function evaluations, often far more than can be performed in practice. As a result, it cannot be used directly, and the evaluation points must instead be carefully selected through other means. Building on the Bayesian optimization and level set estimation frameworks, this is accomplished using two key components: (1) a Bayesian probabilistic model of the function and (2) a sequential sampling policy that uses this model to select new evaluation points. Following [3], we refer to this framework as Bayesian algorithm execution (BAX).

Existing approaches to BAX use expected information gain (EIG) as a criterion for choosing which points to evaluate [3], yet maximizing the EIG poses a significant computational challenge. We propose a simple but effective and scalable algorithm based on posterior sampling to address this challenge. Our approach relies on the key observation that many real-world BAX tasks aim to find a *target set*. For example, in Bayesian optimization, the goal is to find the function's global optimum; in level set estimation, the goal is to find the points above the user-specified threshold. Leveraging this observation, we **propose an algorithm termed PS-BAX** where points are chosen according to the posterior probability of being in the target set. **PS-BAX is scalable and orders of magnitude faster than INFO-BAX**, the EIG-based approach proposed by [3]. Finally, we **prove that PS-BAX is asymptotically consistent** under mild regularity conditions.

Submitted to Workshop on Bayesian Decision-making and Uncertainty, 38th Conference on Neural Information Processing Systems (BDU at NeurIPS 2024). Do not distribute.

**Algorithm 1** PS-BAX

**Require:** $p(f)$ (prior), $\mathcal{D}_0$ (initial dataset), $\mathcal{A}$ (base algorithm), $N$ (number of iterations).
1: **for** $n = 1 : N$ **do**
2:     Sample $\tilde{f}_n$ from $p(f \mid \mathcal{D}_{n-1})$
3:     Apply algorithm $\mathcal{A}$ on $\tilde{f}_n$ to obtain $X_n = \mathcal{O}_{\mathcal{A}}(\tilde{f}_n)$
4:     Choose $x_n \in \text{argmax}_{x \in X_n} \mathbf{H}[f(x)|\mathcal{D}_{n-1}]$     *//Choose $x_n \in X_n$ with highest uncertainty*
5:     Evaluate $y_n = f(x_n)$ and set $\mathcal{D}_n = \mathcal{D}_{n-1} \cup \{(x_n, y_n)\}$
6: **end for**
7: **return** Estimate of $\mathcal{O}_{\mathcal{A}}(f)$ based on $p_N$.     *//E.g., run $\mathcal{A}$ on the final posterior mean*

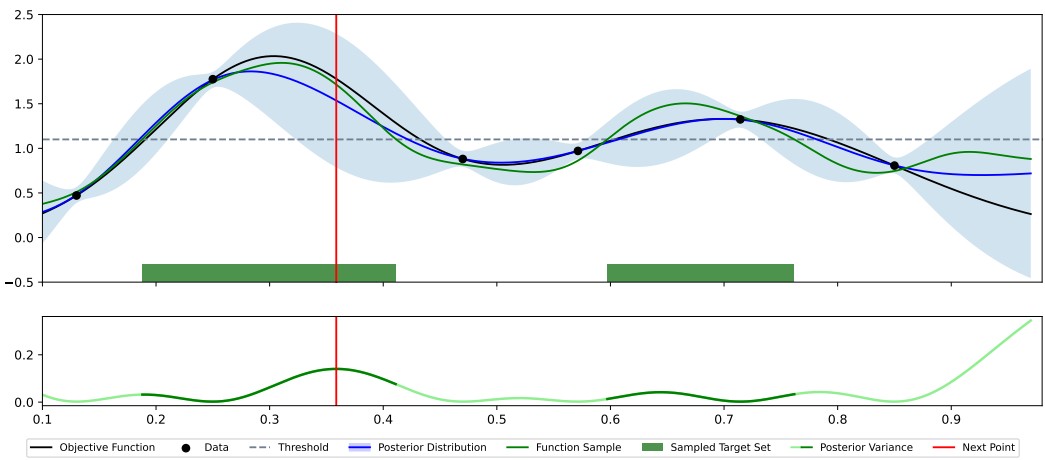

Figure 1: Depiction of PS-BAX (Algorithm 1) for a level-set estimation problem. We plot the objective function $f$ (black line), the available data $\mathcal{D}_n$ (black points), the threshold (grey dashed line), the posterior distribution $p(f \mid \mathcal{D}_n)$ (blue line and light blue region), a sample from the posterior $\tilde{f}_n \sim p(f \mid \mathcal{D}_n)$ (green line), the corresponding sampled target set $X_n = \mathcal{O}_{\mathcal{A}}(\tilde{f}_n)$ (green region) (this is the set of inputs where the green line is above the threshold), the variance of $p(f \mid \mathcal{D}_n)$ (green line, bottom row), and the next point to evaluate selected by PS-BAX $x_{n+1} \in X_n$ (input marked by the vertical red line). The key step is computing the target set $X_n$ using the sampled function $\tilde{f}_n$, which generalizes posterior sampling for Bayesian optimization.

This work is based on our prior work [citation removed to preserve anonymity]. The present version extends our prior work by including new experiments, a generalization of our algorithm to the batch setting, and an improved discussion of theoretical results.

## 2 Bayesian Algorithm Execution via Posterior Sampling

**Problem Setting** Our work takes place within the Bayesian algorithm execution (BAX) framework introduced by [3]. The goal is to estimate $\mathcal{O}_{\mathcal{A}}(f)$, the output of a *base algorithm* $\mathcal{A}$ applied on a function $f : \mathcal{X} \to \mathbb{R}$. We assume that $f$ is expensive to evaluate, which means that employing $\mathcal{A}$ directly on $f$ is infeasible (would require evaluating $f$ too many times). Instead, we will select the points at which $f$ is evaluated sequentially, aided by a probabilistic model. As is standard in the literature, we use Gaussian process models in our experiments. More details are provided in Appendix B However, our framework can easily accommodate other models, provided that sampling from the posterior is feasible. We specifically focus on problems where the property of interest can be encoded by a set $\mathcal{O}_{\mathcal{A}}(f) \subset \mathcal{X}$, which we term the *target set*.

**PS-BAX** Our algorithm, termed PS-BAX, is summarized in Algorithm 1. In words, we first draw a sample from the posterior over $f$, denoted by $\tilde{f}_n$ (line 2), and then set the sample target set $X_n = \mathcal{O}_{\mathcal{A}}(\tilde{f}_n)$. We then select the point in sampled target set $X_n$ with maximal entropy: $x_n \in \text{argmax}_{x \in X_n} \mathbf{H}[f(x)|\mathcal{D}_n]$. For a Gaussian posterior, $x_n$ can be equivalently selected as $x_n \in \text{argmax}_{x \in X_n} \sigma_n(x)$, where $\sigma_n(x)$ is the posterior standard deviation of $f(x)$. Intuitively, PS-

BAX can be seen as a generalization of posterior sampling in Bayesian optimization. However, in general BAX tasks, the target may be comprised of several points; thus, we select the point with the highest *uncertainty* among points in $X_n$, a standard strategy in the active learning literature. The batch generalization of PS-BAX is discussed in Appendix F. A comparison between INFO-BAX and PS-BAX is provided in Appendix D.

**Asymptotic Consistency of PS-BAX**  A natural question is under which conditions is PS-BAX able to *find* the target set given enough evaluations. We provide an answer to this question below. Before formally stating our results, we introduce a definition related to the characterization of problems where PS-BAX is expected to converge.

**Definition 1.** *A target set estimated by an algorithm $\mathcal{A}$ is complement-independent if $\mathcal{O}_{\mathcal{A}}(f) = \mathcal{O}_{\mathcal{A}}(f')$ for and any pair of functions $f$ and $f'$ such that $f(x) = f'(x)$ for all $x \in \mathcal{O}_{\mathcal{A}}(f) \cup \mathcal{O}_{\mathcal{A}}(f')$.*

Many target sets of interest, such as a function's optimum or level set, are complement-independent. Indeed, the value of $f$ at points that are not the optimum or that do not lie in the level of interest do not influence these properties. Theorem 1 below shows that PS-BAX enjoys asymptotic posterior consistency, provided the target set of interest is complement-independent. Intuitively, this result means that if $f$ is drawn from the prior used by our algorithm (i.e., the prior is well-specified), then, with probability one, the posterior will concentrate around the true target set. Corollary 1 gives an asymptotically consistent estimator of the target set. Finally, we also show there are problems where the target set is not complement-independent and PS-BAX is not asymptotically consistent in Theorem 2. The proofs of these results can be found in Appendix C.

**Theorem 1.** *Suppose that $\mathcal{X}$ is finite and that the target set estimated by $\mathcal{A}$ is complement-independent. If the sequence of points $\{x_n\}_n$ is chosen according to the PS-BAX strategy, then, for each $X \subset \mathcal{X}$, $\lim_{n \to \infty} \mathbf{P}_n(\mathcal{O}_{\mathcal{A}}(f) = X) = \mathbf{1}\{\mathcal{O}_{\mathcal{A}}(f) = X\}$ almost surely for $f$ drawn from the prior.*

**Corollary 1.** *Suppose that the assumptions made in Theorem 1 hold and let $T_n \in \mathrm{argmax}_{X \in \mathcal{X}} \mathbf{P}_n(\mathcal{O}_{\mathcal{A}}(f) = X)$. Then, $T_n = \mathcal{O}_{\mathcal{A}}(f)$ for all $n$ large enough almost surely for $f$ drawn from the prior.*

**Theorem 2.** *There exists a problem instance (i.e., $\mathcal{X}$, a Bayesian prior over $f$, and $\mathcal{A}$) such that if the sequence of points $\{x_n\}_n$ is chosen according to the PS-BAX strategy, then there is a set $X \subset \mathcal{X}$ such that $\lim_{n \to \infty} \mathbf{P}_n(\mathcal{O}_{\mathcal{A}}(f) = X) = 1/2$ almost surely for $f$ drawn from the prior.*

## 3 Numerical Experiments

We demonstrate the performance of PS-BAX on four different problem classes, described below below. We compare the performance of PS-BAX against the INFO-BAX [3], and uniform random sampling over $\mathcal{X}$ (Random); when available, we also include an algorithm from the literature specifically designed for the problem class. The performance of each algorithm is determined by running the algorithm $\mathcal{A}$ on $\mu_n$, the posterior mean of $f$ given $\mathcal{D}_n$ and subsequently computing a suitable performance metric on $\mathcal{O}_{\mathcal{A}}(\mu_n)$. Additional details are provided in Appendix E.

(a) **Local Optimization** aims to find the optimum of $f$ using a local optimization method base algorithm (potentially with multiple restarts). In our experiments, we use L-BFGS-B as the base algorithm. The performance metric is the log10 inference regret, given by $\log_{10}(f^* - f(\hat{x}_n^*))$, where $\hat{x}_n^*$ is obtained by applying $\mathcal{A}$ on $\mu_n$. As a baseline, we also include the expected improvement (EI) acquisition function.

(b) **Level Set Estimation** aims to find a $\mathcal{O}_{\mathcal{A}}(f) := \{x \in \mathcal{X} \mid f(x) > \tau\}$ for a user-specified $\tau$. The base algorithm $\mathcal{A}$ is the algorithm that ranks all the objective values and returns the points at which the function value is greater than the threshold. The performance metric we consider is the F1 score. As an additional baseline specifically designed for this setting, we include the LSE algorithm proposed by [2].

(c) **Top-$k$ Estimation** aims to find the $k$ points with the largest values of $f(x)$ on a finite (but potentially large) set $\mathcal{X}$. The base algorithm $\mathcal{A}$ is the algorithm that evaluates $f$ at all points in $\mathcal{X}$ and returns the $k$ best points. We use the Jaccard distance between the estimated output $S_n = \mathcal{O}_{\mathcal{A}}(\mu_n)$ and the ground truth optimal set $S^*$, which is defined as

$$d(S_n, S^*) = 1 - \frac{|S_n \cap S^*|}{|S_n \cup S^*|}. \tag{1}$$

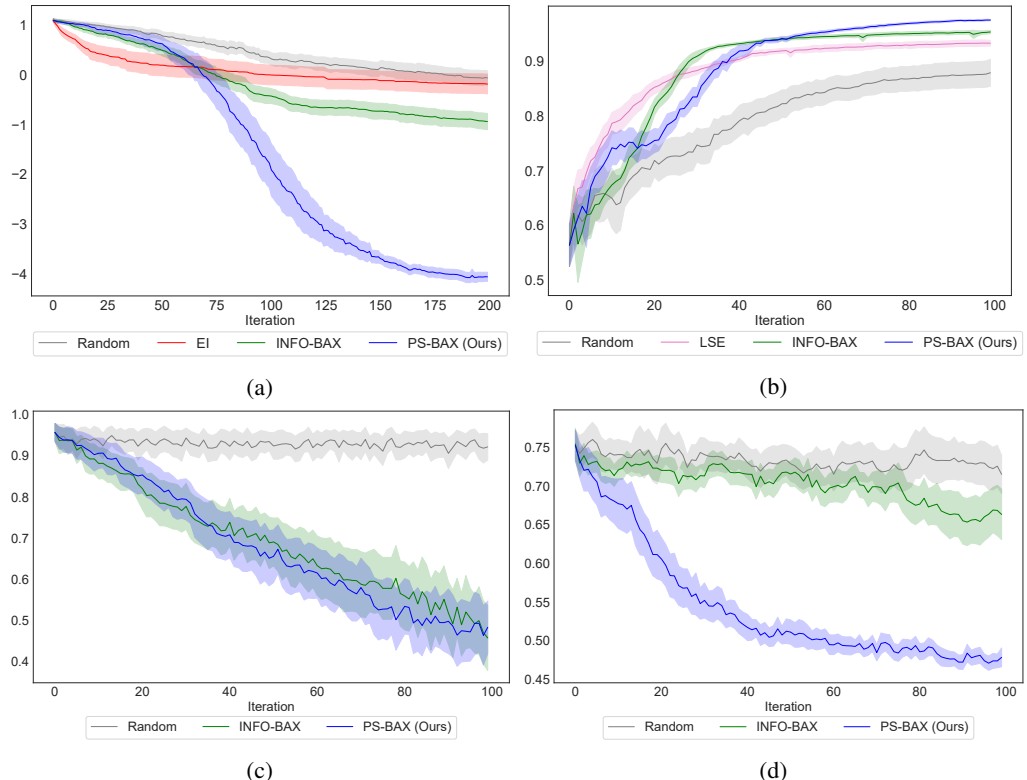

Figure 2: Performance of PS-BAX across four test problems and comparisons against different baselines. (a) The log10 inference regret for the local optimization setting on the Ackley 10D test function. Lower is better. (b) The F1 score for the level set estimation setting on Auckland's Maunga Whau volcano dataset [4] with threshold $\tau = 0.55$ of all the function values in the domain. Higher is better. (c) The Jaccard distance for the top-10 problem on an 80-dimensional domain of size 10000 on the GB1 protein dataset with batch size = 5 [5]. (d) Inference regret for DiscoBAX problem using the Achilles dataset [6], with intervention values from the Interferon $\gamma$ assay [7]. Lower is better.

(d) **DiscoBAX** is a problem setting from [8] where the goal is to find a set of optimal genomic interventions to determine suitable drug targets. Formally, $\mathcal{O}_{\mathcal{A}}$ is the solution of

$$\max_{S \subset \mathcal{X} : |S| = k} \mathbb{E}_\eta \left[ \max_{x \in S} f(x) + \eta(x) \right], \qquad (2)$$

where $\mathcal{X}$ is a pool of genetic interventions, $k$ is the desired interventions set size, $f(x)$ is an *in vitro* measurement correlated to the effectiveness of intervention $x$, and $\eta(x)$ encodes noise and other exogenous factors.

We evaluate the performance of PS-BAX on eight problems across four problem classes (the rest of the experiment results can be. The results for four of the problems (one for each class) are shown in Figure 2. The rest of our experimental results can be found in Appendix E). Overall, we find that PS-BAX is always competitive with and sometimes significantly outperforms INFO-BAX across all of our experiments. Moreover, PS-BAX is one to two orders of magnitude faster in all experiments.

# 4 Conclusion

Many real-world problems can be cast as estimating a property of a black-box function with expensive evaluations. By making the key observation that in many problems, the property of interest is a target set of points defined in terms of the function, we introduce a novel posterior sampling strategy. Our experiments across a broad range of settings show that this strategy is competitive with the approach proposed by [3] while being much faster to compute. Finally, we showed that our posterior sampling strategy is asymptotically consistent under mild regularity conditions.

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

## A  Additional Related Work

Our work falls within the broad field of probabilistic numerics [9], which casts numerical problems, such as optimization or integration, as probabilistic inference problems. This probabilistic approach allows for uncertainty quantification, which is crucial for settings where the computational budget is small and computing must be carefully planned, often in an adaptive fashion. Most of the recent work in probabilistic numerics has focused on (Bayesian) optimization [1, 10]. However, there have also been efforts in integration (Bayesian quadrature) [11–13], level set estimation [2, 14], and solving differential equations [15, 16].

Recently, [3] proposed an approach termed INFO-BAX to estimate an arbitrary property of interest that, in principle, can be computed via a known *base algorithm*. The base algorithm requires a potentially large number of function evaluations and thus cannot be applied directly. Instead, following the probabilistic numerics paradigm, a Bayesian probabilistic model of the function is used to select new points to evaluate iteratively. At each iteration, the point to evaluate next is obtained

by maximizing the expected information gain (EIG) between the point and the property of interest. Similar EIG-based approaches have been used in the statistical design of experiments [17–19] and Bayesian optimization [20–22]. These approaches often deliver an excellent performance. However, they are quite computationally demanding due to the look-ahead nature of the EIG computation. Moreover, in most cases, the EIG cannot be computed in closed form and must be approximated via Monte Carlo sampling. As a consequence, EIG-based approaches are only useful in low-dimensional settings and when the function evaluations are highly expensive, severely limiting their applicability in real-world problems.

In response to the limitations of EIG approaches, we explore an alternative family of strategies known as posterior sampling, or Thompson sampling [23, 24]. Posterior sampling algorithms have been widely used in Bayesian optimization [25–27], multi-armed bandits [28–30], and reinforcement learning [31–33]. In such settings, these approaches select a point at each iteration according to the posterior probability of being the optimum. To our knowledge, our work represents the first extension of posterior sampling beyond optimization settings, offering fresh insights into this algorithmic family. At the same time, we note that the range of problems that can be tackled with our approach is narrower than those that can be tackled, at least conceptually, using the EIG approach. Nevertheless, this class of problems remains substantial. In particular, it encompasses those investigated empirically by [3] and the follow-up work [8], among many others.

Within the optimization setting, our work aligns with recent efforts aimed at broadening the applicability of Bayesian optimization to complex real-world problems. Many such problems depart from classical optimization formulations, exhibiting diverse structures involving combinatorial [8], robust [34, 35], or multi-level optimization formulations [36]. Regular Bayesian optimization algorithms often fail to naturally accommodate these structures, thus limiting their practical utility. Our work introduces a straightforward algorithm applicable to these diverse settings, serving as a robust baseline for future exploration. Finally, our work also benefits from recent advances in probabilistic modeling tools [37–39] and opens the door for the application of these tools in a broader range of problems.

# B  Probabilistic Model

Our algorithm relies on a probabilistic model encoded by a prior distribution over $f$, which we denote by $p_0$. Although our framework is more general and can be used with other priors, we assume for concreteness that $f$ follows a Gaussian process prior [40]. Let $\mathcal{D}_{n-1} = \{(x_k, y_k)\}_{k=1}^{n-1}$ denote the data collected after $n-1$ evaluations of $f$. We assume these evaluations are corrupted with i.i.d. Gaussian noise, i.e., $y_k = f(x_k) + \epsilon_k$, where $\epsilon_1, \ldots, \epsilon_{n-1}$ are i.i.d. with common distribution $\mathcal{N}(0, \sigma^2)$, where $\sigma^2$ is a non-negative scalar. Under these assumptions, the posterior distribution over $f$ given $\mathcal{D}_{n-1}$, denoted by $p_n$, is again a Gaussian process whose mean and covariance functions can be computed in closed form using the classical Gaussian process regression equations.

# C  Proofs of Theorems 1 and 2

## C.1  Proof of Theorem 1

We first introduce the following notation. Let $\mathcal{F}_n$ denote the $\sigma$-algebra generated by $\mathcal{D}_{n-1}$ and $\mathcal{F}_\infty$ denote the minimal $\sigma$-algebra generated by $\{\mathcal{F}_n\}_{n=1}^\infty$. We denote the conditional probability measures induced by $\mathcal{F}_n$ and $\mathcal{F}_\infty$ by $\mathbf{P}_n$ and $\mathbf{P}_\infty$, respectively. In the following results, we assume that the sequence of points $\{x_n\}_{n=1}^\infty$ is selected according to our PS-BAX strategy.

**Lemma 1.** *Suppose that $x \in \mathcal{X}$ is such that $\mathbf{P}_\infty(\mathcal{O}_\mathcal{A}(f) = X) > 0$ for some $X \subset \mathcal{X}$ with $x \in X$. Then, $f(x)$ is $\mathcal{F}_\infty$-measurable.*

*Proof.* A standard martingale argument shows that $\lim_{n \to \infty} \mathbf{P}_n(\mathcal{O}_\mathcal{A}(f) = X) = \mathbf{P}_\infty(\mathcal{O}_\mathcal{A}(f) = X)$. Thus, since $\mathbf{P}_\infty(\mathcal{O}_\mathcal{A}(f) = X) > 0$, it follows that we can find $\epsilon > 0$ such that $\mathbf{P}_n(\mathcal{O}_\mathcal{A}(f) = X) > \epsilon$ for all $n$ large enough. This implies that the event $X_n = X$ occurs infinitely often. Now we consider two cases. If $\sigma_n(x) = 0$ for some $n$, then this necessarily implies that $\mu_n(x) = f(x)$ (see, e.g., [41], Theorem 3.12). If, on the other hand, $\sigma_n(x) > 0$ for all $n$, it is not hard to see that $\sigma_n(x)$ converges to zero if and only if $x$ is selected infinitely often. Since $x_n = \mathrm{argmax}_{x \in X_n} \sigma_n(x)$, it follows that each element in $X$ is selected infinitely often; i.e., the event $x_n = x$ occurs infinitely often. Let

$n_1, n_2, \ldots$ be the sequence of indices such that $x_{n_k} = x$. By the law of large numbers

$$\lim_{K \to \infty} \frac{1}{K} \sum_{k=1}^{K} y_{n,k} = f(x)$$

almost surely. Since $\mu_n(x)$ and $\lim_{K \to \infty} \frac{1}{K} \sum_{k=1}^{K} y_{n,k} = f(x)$ are both $\mathcal{F}_\infty$-measurable, it follows from the analysis of these two cases that $f(x)$ is $\mathcal{F}_\infty$-measurable. □

**Theorem 1.** *Suppose that $\mathcal{X}$ is finite and that the target set estimated by $\mathcal{A}$ is complement-independent. If the sequence of points $\{x_n\}_n$ is chosen according to the PS-BAX strategy, then, for each $X \subset \mathcal{X}$, $\lim_{n \to \infty} \mathbf{P}_n(\mathcal{O}_\mathcal{A}(f) = X) = \mathbf{1}\{\mathcal{O}_\mathcal{A}(f) = X\}$ almost surely for $f$ drawn from the prior.*

*Proof.* A standard martingale argument shows that $\lim_{n \to \infty} \mathbf{P}_n(\mathcal{O}_\mathcal{A}(f) = X) = \mathbf{P}_\infty(\mathcal{O}_\mathcal{A}(f) = X)$ almost surely. Thus, it remains to show that $\mathbf{P}_\infty(\mathcal{O}_\mathcal{A}(f) = X) = \mathbf{1}\{\mathcal{O}_\mathcal{A}(f) = X\}$ almost surely.

Let $Z = \{x \in \mathcal{X} : \mathbf{P}_\infty(\mathcal{O}_\mathcal{A}(f) = X) = 0 \; \forall \, X \subset \mathcal{X} \text{ s.t. } x \in X\}$. By construction, $Z \cap \mathcal{O}_\mathcal{A}(f) = \emptyset$ $\mathbf{P}_\infty$-almost surely. Moreover, from Lemma 1, $f(x)$ is $\mathcal{F}_\infty$-measurable for each $x \in \mathcal{X} \setminus Z$. Since $\mathcal{O}_\mathcal{A}(f)$ is complement-independent, $\mathcal{O}_\mathcal{A}(f)$ is fully determined by the values of $f$ over $\mathcal{X} \setminus Z$. It follows from this that $\mathcal{O}_\mathcal{A}(f)$ is $\mathcal{F}_\infty$-measurable. Hence, $\mathbf{P}_\infty(\mathcal{O}_\mathcal{A}(f) = X) = \mathbf{1}\{\mathcal{O}_\mathcal{A}(f) = X\}$ almost surely under the prior on $f$. □

### C.2   Proof of Theorem 2

**Theorem 2.** *There exists a problem instance (i.e., $\mathcal{X}$, a Bayesian prior over $f$, and $\mathcal{A}$) such that if the sequence of points $\{x_n\}_n$ is chosen according to the PS-BAX strategy, then there is a set $X \subset \mathcal{X}$ such that $\lim_{n \to \infty} \mathbf{P}_n(\mathcal{O}_\mathcal{A}(f) = X) = 1/2$ almost surely for $f$ drawn from the prior.*

*Proof.* Let $\mathcal{X} = \{-1, 0, 1\}$ and consider a GP prior over $f$ such that $f(-1) = f(1) = 0$ and $f(0)$ is a standard normal random variable. Consider the algorithm $\mathcal{A}$ such that $\mathcal{O}_\mathcal{A}(f) = \{-1\}$ if $f(0) < 0$ and $\mathcal{O}_\mathcal{A}(f) = \{1\}$ otherwise. Clearly, the target set obtained from $\mathcal{A}$ is not complement-independent. Moreover, under the PS-BAX strategy, $x_n$ is always either $-1$ or $1$. Since the values of $f$ at these points are known, the posterior distribution over $f$ at any iteration $n$ is equal to the prior. From this it can be easily shown that $\mathbf{P}_n(\mathcal{O}_\mathcal{A}(f) = \{-1\}) = \mathbf{P}_n(\mathcal{O}_\mathcal{A}(f) = \{1\}) = 1/2$ for all $n$. □

# D   Comparison Between INFO-BAX and PS-BAX

## D.1   INFO-BAX and its Shortcomings

Succinctly, the INFO-BAX approach proposed by [3] selects at each iteration the point that maximizes the expected entropy reduction between the function's value at the evaluated point and $\mathcal{O}_\mathcal{A}(f)$. Evaluating an expectation is generally difficult, and one often resorts to Monte Carlo sampling. Moreover, computing the EIG specifically requires expensive calculations of conditional posterior distributions and entropy. These computational issues are also present in similar information-theoretic acquisition functions proposed in the classic Bayesian optimization (BO) setting. However, for BAX tasks, the computation burden of EIG can be much more pronounced if $|\mathcal{O}_\mathcal{A}(f)|$ is large. This occurs, for example, in the level set estimation setting, where $\mathcal{O}_\mathcal{A}(f)$ can be comprised by a large fraction of the entire input space. A more detailed discussion of the computation of the EIG is provided in Section D.2 below.

On the other hand, PS-BAX requires running $\mathcal{A}$ only once on a single sample of $f$, which contributes to of our algorithm's practicality and scalability. Like in posterior sampling for the standard BO setting, our approach sidesteps the need to maximize an acquisition function over $\mathcal{X}$, which is computationally expensive due to needing to compute the expected value of a computationally expensive quantity such as information gain. We refer the reader to Section D.3 for a detailed discussion on the computational complexity of PS-BAX and INFO-BAX.

## D.2   Computation of the Expected Information Gain

Let $\mathbf{E}$ and $\mathbf{H}$ denote the expectation and (differential) entropy operators, respectively. At each iteration $n$, the expected information gain between the $\mathcal{O}_\mathcal{A}(f)$ and a new observation of $f$ at $x$,

denoted by $y_x$, can be written as

$$\text{EIG}_n(x) = \mathbf{H}[y_x \mid \mathcal{D}_n] - \mathbf{E}[\mathbf{H}[y_x \mid \mathcal{D}_n, \mathcal{O}_\mathcal{A}(f)] \mid \mathcal{D}_n]. \tag{3}$$

Under the probabilistic model established above, the conditional distribution of $y_x$ given $\mathcal{D}_n$ is Gaussian, allowing the analytical computation of $\mathbf{H}[y_x \mid \mathcal{D}_n]$. However, in most cases, $\mathbf{H}[y_x \mid \mathcal{D}_n, \mathcal{O}_\mathcal{A}(f)]$ cannot be computed analytically. In particular, this is true in our setting, where $\mathcal{O}_\mathcal{A}(f)$ is a subset of $\mathcal{X}$.

To address this challenge, [3] introduced an approximation where $\mathcal{O}_\mathcal{A}(f)$ is replaced by a small set of pairs $(x', f(x'))$ for inputs $x'$ evaluated when $\mathcal{A}$ is applied on $f$. The corresponding approximation of $\text{EIG}_n$, denoted by $\text{EIG}_n^v$, is then given by

$$\text{EIG}_n^v(x) = \mathbf{H}[y_x \mid \mathcal{D}_n] - \mathbf{E}[\mathbf{H}[y_x \mid \mathcal{D}_n, \{(x', f(x')) : x' \in \mathcal{O}_\mathcal{A}(f)\}] \mid \mathcal{D}_n]. \tag{4}$$

The advantage of this approximation is that, again, $\mathbf{H}[y_x \mid \mathcal{D}_n, \{(x', f(x')) : x' \in \mathcal{O}_\mathcal{A}(f))\}$ can be computed analytically under a Gaussian posterior.

The expectation $\mathbf{E}[\mathbf{H}[y_x \mid \mathcal{D}_n, \{(x', f(x')) : x' \in \mathcal{O}_\mathcal{A}(f)\}] \mid \mathcal{D}_n]$ still requires to be approximated via Monte Carlo sampling. Concretely, this can be achieved by drawing $L$ samples from the posterior over $f$ given $\mathcal{D}_n$, denoted by $\tilde{f}_{n,1}, \ldots, \tilde{f}_{n,L}$, and setting

$$\text{EIG}_n^v(x) \approx \mathbf{H}[y_x \mid \mathcal{D}_n] - \frac{1}{L} \sum_{\ell=1}^{L} \mathbf{H}[y_x \mid \mathcal{D}_n, \{(x', f(x')) : x' \in \mathcal{O}_\mathcal{A}(\tilde{f}_{n,\ell})\}]. \tag{5}$$

This is the approximation of $\text{EIG}_n$ that we use in our experiments in Section 3, i.e., at each iteration, we set $x_n$ to be a point that maximizes the expression in Equation 5. For brevity, we refer to this acquisition function simply as $\text{EIG}_n$.

### D.3 Computational Complexity of INFO-BAX and PS-BAX

Given a Gaussian process posterior, we analyze the complexity of computing the next point to evaluate for PS-BAX and INFO-BAX. Our analysis excludes the cost of generating a sample from the posterior, which is fixed and depends only on the number of Fourier features used. It also assumes that the cost of evaluating such a sample at any given point is 1. Similarly, it assumes that the cost of evaluating the posterior mean and covariance is 1. We further assume that the function domain $\mathcal{X}$ is discrete with $|\mathcal{X}| = N$, the algorithm output has a fixed cardinality $|\mathcal{O}_\mathcal{A}| = M$, the number of execution paths to approximate the posterior entropy is $L$, and running the algorithm on any input function requires $P$ evaluations of the input function. As we shall see, the computational cost of INFO-BAX can be significantly higher than that of PS-BAX if either $N$, $M$, or $L$ is large.

For PS-BAX, the complexity is $O(P + M)$, which represents the complexity of running the algorithm once on one function sample $\tilde{f}$ and maximizing the posterior variance over $\mathcal{O}_\mathcal{A}(\tilde{f})$. For INFO-BAX, the complexity is $O((P + M^3 + N \cdot M^2) \cdot L)$. For each function sample, we need to execute the algorithm ($P$), condition on $M$ new points to find the conditional entropy ($M^3$), and compute the posterior variance of the fantasized model on $N$ points ($N \cdot M^2$). This process is repeated for $L$ function samples.

## E  Additional Details on the Numerical Experiments

### E.1  Implementation Details

In all problems, an initial data set is obtained using $2(d + 1)$ inputs chosen uniformly at random over $\mathcal{X}$, where $d$ is the input dimension of the problem. After this initial stage, each algorithm is used to select additional inputs iteratively. The performance plots show the mean plus and minus two standard errors of the corresponding performance metrics. Each experiment was replicated 30 times. All our algorithms are implemented using BoTorch [37]. In particular, all of our experiments, except for the top-$k$ GB1 protein design task, use BoTorch's `SingleTaskGP` class with its default settings. Approximate samples from the posterior on $f$ used by both PS-BAX and INFO-BAX are obtained using 1000 random Fourier features [42]. Our implementations of both PS-BAX and INFO-BAX provide automatic computation of gradients, which allows continuous optimization when $\mathcal{X}$ is continuous. For INFO-BAX, we set the number of Monte Carlo samples to estimate the EIG equal to $L = 30$ across all problems.

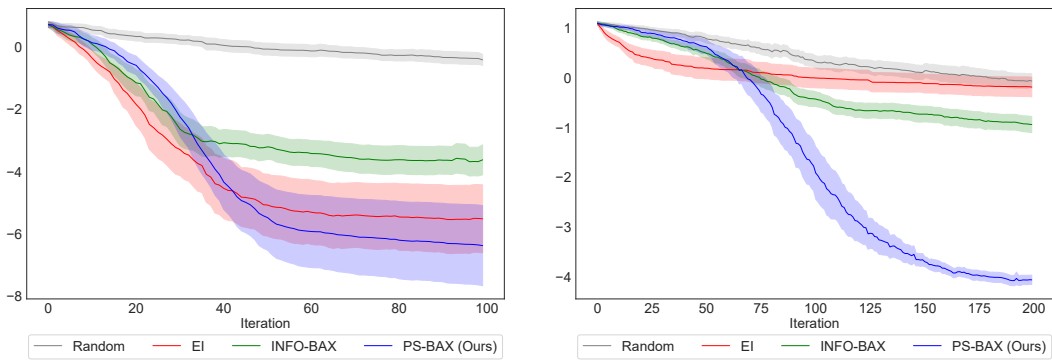

Figure 3: Results for the local optimization setting showing the log10 inference regret achieved by the compared algorithms. The left and right panels show results for the Hartmann-6D, Ackley-10D functions, respectively. PS-BAX and EI are comparable on Hartmann-6D, both surpassing INFO-BAX. On Ackley-10D, PS-BAX is significantly better. Lower is better.

## E.2 Local Optimization

We explore the performance of our algorithm in the local optimization setting, where $\mathcal{A}$ is a local optimization algorithm, assuming that $f$ (and potentially its gradients) can be evaluated at a large number of points. Examples of such algorithms include evolutionary algorithms [43], trust-region methods [44], and many gradient-based optimization algorithms [45–47]. This setting reduces to the classical BO setting if $\mathcal{A}$ can recover the global optimum of $f$. In such case, the INFO-BAX reduces to the classical predictive entropy search acquisition function [21] when computed exactly and to the joint entropy search acquisition function [48] under the approximation proposed by [3] we use in our experiments. PS-BAX, in turn, reduces to the classical posterior sampling strategy used in BO [25].

In our experiments, we use a gradient-based optimization method as a base algorithm instead of an evolutionary algorithm as pursued by [3]. Gradient-based methods typically exhibit faster convergence than their gradient-free counterparts. However, they are often infeasible if gradients cannot be obtained analytically and instead are obtained, e.g., via finite differences. Since in most applications, analytic gradients of $f$ are unavailable, directly applying such methods on $f$ is infeasible. However, INFO-BAX and PS-BAX can make use of gradient-based methods thanks to the availability of gradients of most probabilistic models used in practice, including Gaussian processes.

We consider the Hartmann and Ackley functions, with input dimensions of 6 and 10, respectively, as test functions. Both functions have many local minima and are standard test functions in the BO literature. As a performance metric, we report the log10 inference regret, given by $\log_{10}(f^* - f(\hat{x}_n^*))$, where $\hat{x}_n^*$ is obtained by applying $\mathcal{A}$ on $\mu_n$. The results of these experiments are depicted in Figure 3. As a baseline, we also include the expected improvement (EI).

## E.3 Level Set Estimation

Level set estimation is the task of finding points in $\mathcal{X}$ for which $f(x) > \tau$ for a user-specified value of $\tau$. Such tasks arise in environmental monitoring applications, where a mobile sensing device takes measurements to detect regions with dangerous pollution levels [2], and topographic applications, where the goal is to infer the portion of a large geographic region above a specified altitude using a small number of measurements [4]. The base algorithm $\mathcal{A}$ in this case is simply the algorithm that ranks all the objective values and returns the points at which the function value is greater than the threshold. We evaluate our algorithm and benchmarks on both a synthetic problem (the 2 dimensional Himmelblau function) and the Auckland's Maunga Whau volcano dataset [4], constituted by $87 \times 61$ height measurements in a large geographic area around the volcano. The threshold $\tau$ is set to be the $0.55$ quantile of all the function values in the domain for both problems. An illustration of running INFO-BAX and our PS-BAX is shown in Figure 4. The performance metric we consider is the F1 score, given by $F1 = 2TP/(2TP + FP + FN)$, where $TP$ is the number of true positives, $FP$ is the number of false positives, and $FN$ is the number of false negatives.

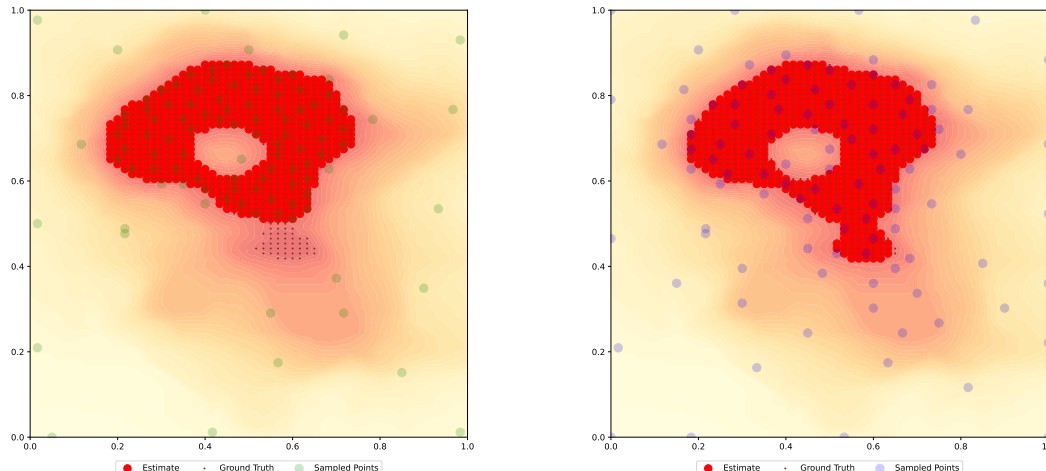

Figure 4: Example of using the INFO-BAX (left) and PS-BAX (right) policy on volcano level-set estimation problem described in Section E.3. The figures show the algorithm output by running the algorithm on the posterior mean, the ground truth super-level-set, and all the points evaluated by each algorithm after 100 iterations. PS-BAX provides an accurate estimate of the level set, whereas INFO-BAX misses a significant portion.

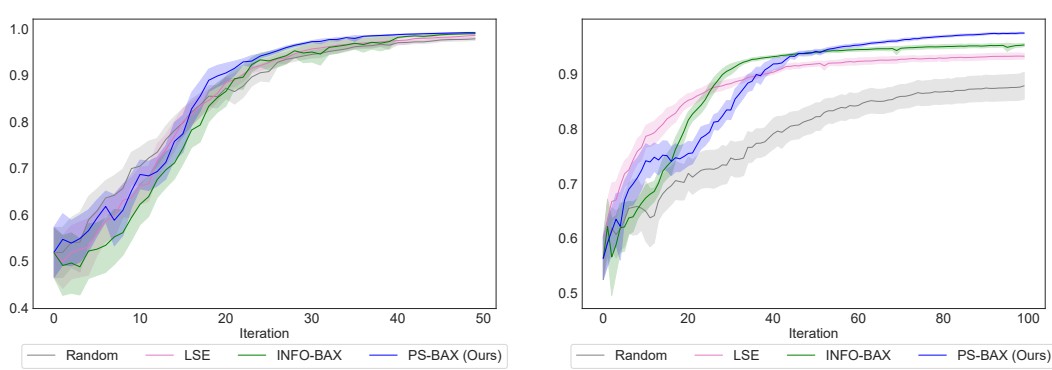

Figure 5: Comparison for the level-set estimation problem. (Left) Results for the 2-dimensional Himmelblau function. (Right) Results for the Maunga Whau volcano dataset.

The results of this experiment are depicted in Figure 5. As an additional baseline specifically designed for level set estimation, we include the popular LSE algorithm proposed by [2]. PS-BAX again exhibits a strong performance, surpassing all the benchmarks.

### E.4 Top-$k$ Estimation

We consider the top-$k$ estimation setting where $\mathcal{X}$ is a finite set, and our goal is to find the $k$ points with the largest values of $f(x)$. As a base algorithm, we use the algorithm that simply evaluates $f$ at all points in $\mathcal{X}$ and returns the $k$ best points. Following [3], we use the Jaccard distance between the estimated output $S_n = \mathcal{O}_{\mathcal{A}}(\mu_n)$ and the ground truth optimal set $S^*$, which is defined as

$$d(S_n, S^*) = 1 - \frac{|S_n \cap S^*|}{|S_n \cup S^*|}. \tag{6}$$

**Synthetic Function** We use the 3-dimensional Rosenbrock test function, which is a standard benchmark in the literature. The input space is obtained by taking a uniform grid over the original input spaces of this function.

**GB1 Protein Design** We also consider a real-world top-$k$ selection problem in the realm of protein design, where the task is to maximize stability fitness predictions for the Guanine nucleotide-binding

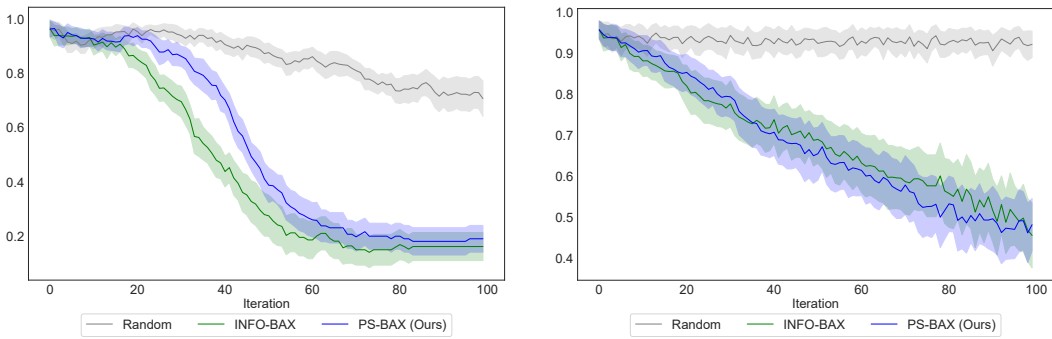

Figure 6: Comparison for the top-$k$ problem using the Jaccard difference metric. Lower is better. (Left) Finding the top 6 points on a 3-dimensional domain with 1000 candidate points in the discretized set using the Rosenbrock function with batch size = 1. (Right) Finding the top 10 points on a 80-dimensional action space of size 10000 on the GB1 dataset with batch size = 5.

protein GB1 given different sequence mutations in a target region of 4 residues [5]. There are $20^4$ possible orderings given 20 amino acids and four positions. GB1 has been well studied by biologists, and its domain is known to be highly rugged and dominated by "dead" variants with very low fitness scores [49]. Due to its high input dimensionality, enormous input space, and sparse fitness landscape, this dataset is very challenging for standard GP models, and thus, we utilize Deep Kernel Learning as proposed in [50] as our probabilistic model. As the dataset size is large, we perform batched evaluations (with a batch size of 5) for PS-BAX and INFO-BAX. The description of the batch extensions of PS-BAX and INFO-BAX can be found in Appendix F.

### E.5 DiscoBAX: Drug Discovery Application

As a final application, we consider the DiscoBAX problem setting from [8], where the task is to find a set of optimal genomic interventions to determine suitable drug targets. Formally, let $\mathcal{X}$ denote a pool of genetic interventions and for each $x \in \mathcal{X}$ let $f(x)$ be an *in vitro* phenotype measurement correlated to the effectiveness of genetic intervention $x$. It is assumed that actual effectiveness of the genomic intervention is not $f(x)$ itself but rather $f(x) + \eta(x)$, where $\eta(x)$ encodes noise and other exogenous factors not captured by the *in vitro* measurement. Following the settings in [8], we simulate $\eta$ using a Gaussian process with mean $0$ and an RBF covariance function. The goal is to find a small set of genomic interventions in $\mathcal{X}$ that maximize an objective function embodying two goals: high expected change in the target phenotype and high diversity to maximize chances of success in the following stages of drug development. The work of [8] formalizes this by introducing the combinatorial optimization problem

$$\max_{S \subset \mathcal{X}:|S|=k} \mathbb{E}_\eta \left[ \max_{x \in S} f(x) + \eta(x) \right], \tag{7}$$

where $k$ is the desired set of interventions. Solving Equation 7 is challenging due to its combinatorial structure. However, a computationally efficient approximation can be obtained by leveraging the submodularity of the objective function. We refer the reader to [8] for more details.

Following [8], we use the Achilles dataset. The gene embeddings of this dataset are represented by 808-dimensional vectors. However, we perform Principal Component Analysis (PCA) as a dimensionality reduction mechanism and then fit a GP to the lower dimensional representation. In addition, we truncate the dataset to 5000 genes with the highest intervention values to keep the runtime of INFO-BAX computationally feasible. The results are shown in Figure 7. PS-BAX significantly outperforms INFO-BAX, whose performance is barely better than that of Random.

## F   Batch Extensions of PS-BAX and INFO-BAX

In this section, we discuss extensions of the PS-BAX and INFO-BAX algorithms to the batch setting, where at each iteration, we generate $q$ new points to evaluate, denoted by $x_n^1, \cdots, x_n^q$. These extensions are inspired by batch extensions of the posterior sampling [25] and joint entropy search

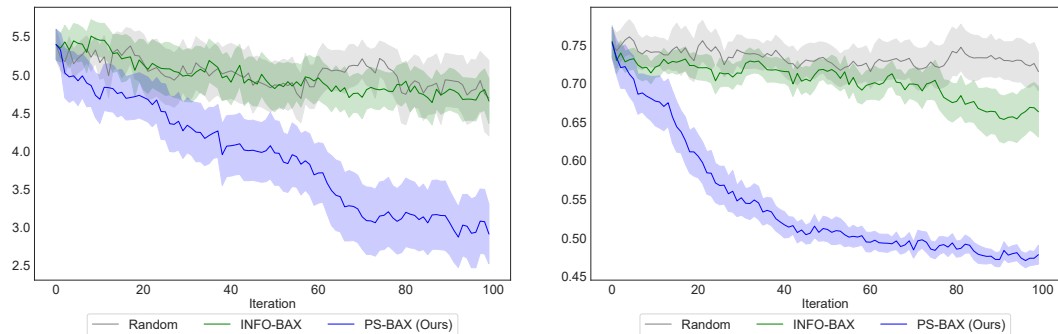

Figure 7: Comparison of DiscoBAX [8] with batch size = 1 on the Achilles dataset [6], with interventions from the Tau protein assay in [7] (left) and the Intergeron $\gamma$ assay (right). The metric reported is the regret, which is the difference between the objective (Eq. 7) values of the algorithm output (the select set of genes) on the posterior mean and the optimum value. Lower is better.

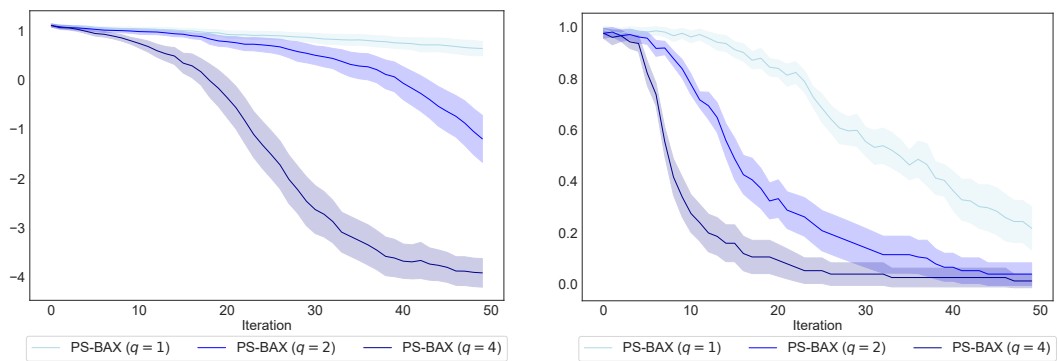

Figure 8: Performance of PS-BAX under various batch sizes (q = 1, 2, 4) on the local optimization Ackley 10-D test problem (left) and the Top-$4$ Himmelblau problem.

acquisition functions [48]. Figure 8 shows the performance of PS-BAX under various batch sizes in two of our test problems.

**Batch PS-BAX** To enable parallel evaluations, the batched version of PS-BAX is implemented as follows. For a batch size of $q$, we draw $q$ samples from the posterior on $f$, denoted by $\tilde{f}_1, \cdots, \tilde{f}_q$ and set $\mathcal{T} = \cup_{i=1}^{q} \mathcal{O}_{\mathcal{A}}(\tilde{f}_i)$. We then select $x_n^1, \cdots, x_n^q$ iteratively by maximizing the joint posterior entropy of these points, i.e.,

$$x_n^1 = \text{argmax}_{x \in \mathcal{S}} \ \mathbf{H}[f(x) \mid \mathcal{D}_n],$$
$$\vdots$$
$$x_n^q = \text{argmax}_{x \in \mathcal{S}} \ \mathbf{H}\big[f(x) \mid \mathcal{D}_n \cup \{x_n^1, \cdots, x_n^{q-1}\}\big]$$

**Batch INFO-BAX** INFO-BAX can be naturally generalized to the batch setting by considering the EIG over a batch of $q$ points. Directly optimizing the EIG, in this case, requires optimizing over $\mathcal{X}^q$, which may be too challenging. Instead we pursue a greedy optimization approach that leverages the submodularity of the EIG. Specifically, we select $x_n^1, \cdots, x_n^q$ iteratively by maximizing the joint posterior entropy of these points, i.e.,

$$x_n^1 = \text{argmax}_{x \in \mathcal{X}} \mathbf{H}[y_x \mid \mathcal{D}_n] - \mathbf{E}[\mathbf{H}[y_x \mid \mathcal{D}_n, \mathcal{O}_{\mathcal{A}}(f)] \mid \mathcal{D}_n],$$
$$\vdots$$
$$x_n^q = \text{argmax}_{x \in \mathcal{X}} \mathbf{H}\big[y_x \mid \mathcal{D}_n \cup \{x_n^1, \cdots, x_n^{q-1}\}\big]$$
$$- \mathbf{E}\big[\mathbf{H}\big[y_x \mid \mathcal{D}_n \cup \{x_n^1, \cdots, x_n^{q-1}\}, \mathcal{O}_{\mathcal{A}}(f)\big] \mid \mathcal{D}_n \cup \{x_n^1, \cdots, x_n^{q-1}\}\big].$$

