# OpenReview forum: "Practical Bayesian Algorithm Execution via Posterior Sampling"
_NeurIPS.cc/2024/Workshop/BDU — NeurIPS BDU Workshop 2024 Poster_

### Official Review · Reviewer_452a · 2024-09-25

**Rating:** 5
**Confidence:** 3

**Review:**

The paper is not very well-written and needs further improvement for clarity.  For example:

- Some grammar problems in Definition 1:  "for and any" -> "for any"?
- incomplete sentence: "the rest of the experiment results can be."
- "PS-BAX is one to two orders of magnitude faster in all experiments" -> faster in what? wall-clock runtime or the number of evaluations?
-  Theorem 2 is more of a specific instance rather than a "theorem"?

PS-BAX seems to be the same as uncertainty sampling (with a constrained search space). It would be good for the authors to make more clarifications/connections.

I am not sure if a local optimization experiment on a single target function for showing the superiorness of PS-BAX is necessary and appropriate. PS-BAX becomes Thompson sampling in this case. Again, it would be good to elaborate more on the connections/relationship with the other methods, instead of just showing "OK it performs great on several problems". Further, does EI mean Bayesian optimization with EI? And what does "Random" mean here?

Overall, though simple and intuitive, the method is an interesting extension of the BAX framework and may be useful and more efficient in certain cases, but I feel uncomfortable that the authors don't elaborate more explicitly on the connections with existing works (maybe partially for supporting the novelty of the algorithm and new theorem proofs). I would suggest the authors to further improve the paper.

---

### Official Review · Reviewer_SxPL · 2024-09-25
**PS-BAX: a new posterior sampling algorithm based on the Bayesian framework and applied to many real-world problems**

**Rating:** 8
**Confidence:** 4

**Review:**

This paper presents a new posterior sampling method entitled PS-BAX. Points are sampled by maximizing the entropy. This work is based in part on the Info-BAX method published in 2021. A concise and clear comparison is made between the two methods. PS-Bax is much faster, its complexity is much lower than Info-Bax and the method seems more robust. \
 The paper is clearly written and pleasant to read. The bibliographical work and the contextualization and perspective are very well done. The paper combines theoretical proofs with a series of benchmarks based on real-world problems. In addition, other methods are added to Info-Bax. \
The paper could be further improved by making figure 2 clearer. I think it would be a good idea to include legends on the Y axis of the figures, and to remind readers of the meaning of the acronyms EI, LSE in the figure text.

---

### Decision · Program_Chairs · 2024-10-09

Accept (Poster)